Placental vascular pathology and increased thrombin generation as mechanisms of disease in obstetrical syndromes

Mastrolia Salvatore Andrea 1 2
Mazor Moshe 2
Loverro Giuseppe 1
Klaitman Vered 2
Erez Offer 2 erezof@bgu.ac.il
1 Department of Obstetrics and Gynecology, Azienda Ospedaliera-Universitaria Policlinico di Bari, School of Medicine, University of Bari “Aldo Moro” , Bari , Italy
2 Department of Obstetrics and Gynecology, Soroka University Medical Center, School of Medicine, Ben Gurion University of the Negev , Beer Sheva , Israel
Feys Hendrik
Electronic publication date: 2014 Nov 18
Publication date: 2014
Volume: 2
Electronic Location ID: e653
Received 2014 Sep 3; Accepted 2014 Oct 14
Copyright: © 2014 Mastrolia et al.
Copyright year: 2014
Copyright holder: Mastrolia et al.
License: This is an open access article distributed under the terms of the Creative Commons Attribution License, which permits unrestricted use, distribution, reproduction and adaptation in any medium and for any purpose provided that it is properly attributed. For attribution, the original author(s), title, publication source (PeerJ) and either DOI or URL of the article must be cited.
License URL: https://creativecommons.org/licenses/by/4.0/

Keywords: Coagulation, TAT III complexes, TFPI, Protein Z, Amniotic fluid, IUGR, PROM, Preeclampsia, Fetal demise, Preterm labor

Funding: The authors declare there was no funding for this work.

==============================
Obstetrical complications including preeclampsia, fetal growth restriction, preterm labor, preterm prelabor rupture of membranes and fetal demise are all the clinical endpoint of several underlying mechanisms (i.e., infection, inflammation, thrombosis, endocrine disorder, immunologic rejection, genetic, and environmental), therefore, they may be regarded as syndromes. Placental vascular pathology and increased thrombin generation were reported in all of these obstetrical syndromes. Moreover, elevated concentrations of thrombin-anti thrombin III complexes and changes in the coagulation as well as anticoagulation factors can be detected in the maternal circulation prior to the clinical development of the disease in some of these syndromes. In this review, we will assess the changes in the hemostatic system during normal and complicated pregnancy in maternal blood, maternal–fetal interface and amniotic fluid, and describe the contribution of thrombosis and vascular pathology to the development of the great obstetrical syndromes.

Introduction

Obstetrical complications including preeclampsia, fetal growth restriction, preterm labor, preterm prelabor rupture of membranes and fetal demise are all the clinical endpoint of several underlying mechanisms (i.e., infection, inflammation, thrombosis, endocrine disorder, immunologic rejection, genetic, and environmental), therefore, they may be regarded as syndromes. In this review, we will assess the changes in the hemostatic system during normal and complicated pregnancy in maternal blood, maternal–fetal interface and amniotic fluid, and describe the contribution of thrombosis and vascular pathology to the development of the great obstetrical syndromes.

What are the Great Obstetrical Syndromes?

The major obstetrical complications including preeclampsia, intrauterine growth restriction (IUGR), preterm labor (PTL), preterm prelabor rupture of membranes (PROM), fetal demise, and recurrent abortions are all syndromes, also defined as “great obstetrical syndromes”. As reported in The Oxford Medical Dictionary a syndrome is ‘a combination of symptoms and/or signs that form a distinct clinical picture indicative of a particular disorder’. Hence, they represent the clinical manifestation of many possible underlying mechanisms of disease (Concise Medical Dictionary, 2010).

Key features of these syndromes are (Romero, 2009): multiple etiologies; long preclinical stage; frequent fetal involvement; clinical manifestations which are often adaptive in nature; and predisposition to a particular syndrome is influenced by gene–environment interaction and/or complex gene–gene interactions involving maternal and/or fetal genotypes. These mechanisms of disease were identified and reported in all the obstetrical complications listed above. This review is focused on the role of thrombosis and vascular pathology of the placenta in these syndromes.

What are the Changes in the Coagulation System During Normal Pregnancy?

In term of the coagulation and hemostatic systems there are several major compartments: the maternal circulation, the fetal maternal interface (the placenta, and the membranes), the amniotic fluid and the fetus and each has a specific behavior during gestation. The changes in the coagulation system during gestation are considered to be adaptive mechanisms and are aimed to: (1) the prevention of bleeding at the time of trophoblast implantation and the delivery of the fetus; (2) allow the laminar flow at the intervillous space; and (3) seal amniotic fluid leak and reduce obstetrical bleeding (Bellart et al., 1998; Walker et al., 1997; Sørensen, Secher & Jespersen, 1995; Yuen, Yin & Lao, 1989; de Boer et al., 1989). Of interest, the fetus is somewhat less involved and its coagulation system develops during gestation, and this subject is beyond the scope of this review.

Indeed, normal pregnancy has been associated with excessive maternal thrombin generation (Bellart et al., 1998; Chaiworapongsa et al., 2002) and a tendency for platelets to aggregate in response to agonists (Yoneyama et al., 2000; Sheu et al., 2002). Pregnancy is accompanied by 2 to 3-fold increase in fibrinogen concentrations and 20% to 1000% increase in factors VII, VIII, IX, X, and XII, all of which peak at term (Bremme, 2003). The concentration of vWF increase up to 400% by term (Bremme, 2003). By contrast, those of pro-thrombin and factor V remain unchanged while the concentrations of factors XIII and XI decline modestly (Eichinger et al., 1999). Indeed there is evidence of chronic low-level thrombin and fibrin generation throughout normal pregnancy as indicated by enhanced concentrations of pro-thrombin fragment 1 and 2, thrombin–antithrombin (TAT) III complexes, and soluble fibrin polymers (Ku et al., 2003). Free protein S concentration declines significantly (up to 55%) during pregnancy due to increased circulating complement 4B-binding protein, its molecular carrier. Protein S nadir at delivery and this reduction is exacerbated by cesarean delivery and infection (Bremme, 2003; Eichinger et al., 1999). As a consequence, pregnancy is associated with an increase in resistance to activated protein C (Eichinger et al., 1999; Ku et al., 2003). The concentration of PAI-1 increase by 3 to 4-fold during pregnancy while plasma PAI-2 values, which are negligible before pregnancy, reach concentrations of 160 mg/L at delivery (Bremme, 2003). Thus, pregnancy is associated with increased clotting potential, as well as decreased anticoagulant properties, and fibrinolysis (Lockwood, 2006). Therefore, it can be defined as a prothrombotic state. One of the most important mediators of the hypercoagulable state of normal pregnancy is tissue factor (TF). Indeed, there is a substantial increase in TF concentrations in the decidua and myometrium (Erlich et al., 1999; Kuczyński et al., 2002; Lockwood, Krikun & Schatz, 2001; Lockwood, Krikun & Schatz, 1999), preventing placental abruption since this leads to an increase in the efficiency of clotting function (Lockwood, 2006). The placenta is a source of TF, since trophoblast cells constitutively express it, behaving as activated endothelium, and leading to a condition of procoagulant state that, if not controlled by anticoagulant mechanisms, predisposes to thrombotic complications (Erlich et al., 1999). The principal anticoagulant mechanism inhibiting TF activation pathway is tissue factor pathway inhibitor (TFPI), which mRNA is highly expressed in the macrophages in the villi in term placenta (Edstrom, Calhoun & Christensen, 2000).

Similarly, high TF concentrations have been detected in the fetal membranes (mainly the amnion) and amniotic fluid (de Boer et al., 1989; Uszyński et al., 2001; Lockwood et al., 1991; Omsjø et al., 1985; Creter, 1977). TFPI has been found in amniotic fluid as well (Uszyński et al., 2001), but it is not clear if the presence of TF and its natural inhibitor is related to coagulation per se or is somehow connected with embryonic development (Carmeliet et al., 1996).

In contrast to the changes detected in the amniotic fluid and the decidua, the median maternal plasma immunoreactive TF concentration of normal pregnant women do not differ significantly from that of non-pregnant patients (Bellart et al., 1998; Holmes & Wallace, 2005). However, labor at term increases significantly the maternal plasma immunoreactive TF concentration in comparison to the non-pregnant state (Uszyński et al., 2001). In addition to the changes in TF, normal pregnancy is associated with increased thrombin generation (Bellart et al., 1998; Chaiworapongsa et al., 2002), as determined by the elevation of maternal concentrations of fibrinopeptide A, prothrombin fragments (PF) 1 and 2, and thrombin–antithrombin (TAT) III complexes (de Boer et al., 1989; Reber, Amiral & de Moerloose, 1998; Uszyński, 1997; Reinthaller, Mursch-Edlmayr & Tatra, 1990). The concentration of these complexes further increases during and after normal parturition (Uszyński, 1997; Andersson et al., 1996), and subsequently decreases during the puerperium (Uszyński, 1997; Andersson et al., 1996).

What are the Changes in the Hemostatic System Associated with the Great Obstetrical Syndromes?

The great obstetrical syndromes are associated with changes in the hemostatic and vascular systems in the compartments mentioned above: (1) the maternal circulation; (2) the feto-maternal interface of placenta and membranes; (3) and the amniotic fluid.

Changes in the hemostatic system of women with obstetrical syndromes

The involvement of the hemostatic system in the pathophysiology of these obstetrical syndromes is becoming more and more apparent. Indeed, increased thrombin generation is reported in the maternal circulation of women with preeclampsia (Schjetlein et al., 1999; Chaiworapongsa et al., 2002; Hayashi et al., 1998; Kobayashi & Terao, 1987; Hayashi et al., 2002), IUGR (Schjetlein et al., 1999; Chaiworapongsa et al., 2002; Hayashi et al., 1998; Hayashi & Ohkura, 2002; Ballard & Marcus, 1972), fetal demise (Erez et al., 2009), PTL (Chaiworapongsa et al., 2002; Erez et al., 2009; Elovitz, Baron & Phillippe, 2001) and preterm PROM (Chaiworapongsa et al., 2002; Erez et al., 2009; Rosen et al., 2001).

There are several possible explanations for the increased thrombin generation in these patients: (1) increased activation of coagulation cascade in the maternal circulation due to pathological processes including bleeding or inflammation; and (2) depletion of anticoagulation proteins that subsequently leads to increased thrombin generation (Table 1).

Table 1 Concentration and activity in maternal plasma of coagulating and anticoagulating factors and their relation with thrombin generation in the great obstetrical syndromes.

	TF concentration
and/or activity	TFPI concentration
and/or activity	TAT III complexes
concentration	Protein Z
concentration	Thrombin
generation	References	
Premature
rupture of membranes	Activity ↑
Concentration ↑	Concentration ↓	Concentration ↑	Concentration ↓	↑	(Erez et al., 2010; Erez et al., 2009; Kusanovic et al., 2007; Gris et al., 2002; Paidas et al., 2005)	
Preterm labor	Activity ↑
Concentration =	Activity =
Concentration ↓	Concentration ↑	Concentration ↓	↑	(Erez et al., 2010; Erez et al., 2009; Kusanovic et al., 2007; Gris et al., 2002; Paidas et al., 2005)	
Fetal demise	Activity =
Concentration =	Activity =
Concentration ↓	Concentration ↑	Concentration ↓	↑	(Erez et al., 2010; Erez et al., 2009; Kusanovic et al., 2007; Gris et al., 2002; Paidas et al., 2005)	
Preeclampsia	Activity ↑
Concentration ↑	Concentration ↑	Concentration ↑	Concentration ↓	↑	(Erez et al., 2010; Erez et al., 2009; Kusanovic et al., 2007; Gris et al., 2002; Paidas et al., 2005)	
Intrauterine growth
retardation/small
for gestational age	Activity ↑
Concentration ↓	Concentration =	Concentration ↑	Concentration ↓	↑	(Erez et al., 2010; Erez et al., 2009; Kusanovic et al., 2007; Gris et al., 2002; Paidas et al., 2005)	

Increased activation of the coagulation cascade and thrombin generation in the maternal circulation in patients with pregnancy complications

All the obstetrical syndromes including preeclampsia (Schjetlein et al., 1999; Chaiworapongsa et al., 2002; Hayashi et al., 1998; Kobayashi & Terao, 1987; Hayashi et al., 2002; Kobayashi et al., 1999; Kobayashi et al., 2002), IUGR (Chaiworapongsa et al., 2002; Hayashi et al., 1998; Hayashi & Ohkura, 2002; Ballard & Marcus, 1972), fetal demise (Erez et al., 2009), PTL (Chaiworapongsa et al., 2002; Elovitz, Baron & Phillippe, 2001) and preterm PROM (Chaiworapongsa et al., 2002; Erez et al., 2009; Rosen et al., 2001) are associated with a higher maternal thrombin generation than a normal pregnancy. These may be of clinical implication since, in women with preterm labor, elevated maternal plasma TAT III complex concentration was associated with a higher chance to deliver within <7 days from admission (Erez et al., 2009) (Fig. 1). To further understand how thrombin affects the duration of pregnancy and the clinical phenotype of patients with the obstetrical syndromes we need to consider what mechanisms lead to thrombin generation and how it affects the feto-maternal unit.

Figure 1 Thrombin–antithrombin (TAT) III levels in control patients, patients with preterm labor who delivered between 21 and 7 days, and patients with preterm labor who delivered within 7 days from admission.

Open diamonds, mean levels; black error bars, SD. ∗P < .05, Student-Newman-Keuls method. From Elovitz, Baron & Phillippe (2001).

Increased thrombin generation can result from the following underlying mechanisms: (1) decidual hemorrhage that leads to a retro-placental clot formation (Lockwood et al., 2005); (2) intra-amniotic infection/inflammation which can induce decidual bleeding and sub-clinical abruption (Gómez et al., 2005), as well as increased intra-amniotic TAT complexes (Erez et al., 2009); and (3) an increased maternal systemic inflammatory response (Gervasi et al., 2002) that may activate the extrinsic pathway of coagulation due to the expression and release of TF by activated monocytes (Ø sterud & Bjørklid, 2006).

Thrombin affects many systems including also the following: (1) stimulation of decidual cell secretion of matrix metalloproteinases (MMP) (i.e., MMP-1 and MMP-3) that can degrade the extracellular matrix of the chorioamniotic membranes (Rosen et al., 2002; Mackenzie et al., 2004) (as in preterm PROM); (2) myometrial activation and uterine contractions generation that may lead to preterm labor with or without rupture of membranes and a subsequent preterm delivery (Elovitz, Baron & Phillippe, 2001; Elovitz et al., 2000a; Elovitz et al., 2000b); and (3) thrombin has an inhibitory effect on the production of TFPI by endothelial cells (Bilsel et al., 2000), and the increased thrombin generation observed in patients with PTL may be associated with a concomitant reduction in TFPI production by the maternal vascular endothelium (the depletion of anticoagulant proteins will be discussed in the following section of this review).

There is evidence to support that the extrinsic pathway of coagulation is activated in many of these pregnancy complications and it is the source of the increased thrombin generation (VanWijk et al., 2002). Indeed, increased immunoreactive TF concentrations were reported in women with preeclampsia and those with preterm PROM (Erez et al., 2010). Moreover, the contribution of preeclampsia to elevated maternal immunoreactive TF persisted also among patients with fetal demise, while those with fetal death who were normotensive did not have higher median TF concentration than normal pregnant women. Indeed, the median TF concentration of patients with preeclampsia was higher than in patients with fetal demise without hypertension. These findings are consistent with previous studies (Bellart et al., 1998; Erez et al., 2008), suggesting that elevated TF immunoreactivity and activity may be associated with the pathophysiologic process leading to preeclampsia, rather than being a consequence of the fetal death.

In some of the obstetrical syndromes there was elevated TF activity in the maternal circulation without a concomitant increase in the plasma concentration of this factor. This was the case among patients with a small for gestational age (SGA) neonate and those with preterm labor (Chaiworapongsa et al., 2002; Erez et al., 2009) (Table 1). This suggests that the increased TF activity among patients with PTL as well as those with an SGA neonate, contributes to a higher generation of factor Xa that, along with the physiologic increase in the maternal plasma concentrations of factor VII and factor X during gestation (Bremme, 2003; Beller & Ebert, 1982; Stirling et al., 1984; Brenner, 2004), may be the underlying mechanism leading to the increased thrombin generation reported these syndromes.

The differences between PTL and preterm PROM in term of maternal plasma TF concentration and activity may derive from the specific component of the common pathway of parturition, which is activated in each obstetrical syndrome (Romero et al., 2006). While preterm PROM is associated with the activation of the decidua and the membranes, myometrial activation is the major component of preterm labor with intact membranes (Romero et al., 2006). This is relevant because the decidua and the membranes have a high TF concentration (Lockwood, Krikun & Schatz, 2001; Lockwood, Krikun & Schatz, 1999; Lockwood et al., 2007).

In summary, the evidence brought herein suggests that increased thrombin generation in patients with the great obstetrical syndromes may reflect the activation of the coagulation cascade mainly through the extrinsic arm. This activation can be attributed to various underlying mechanisms.

Depleted or insufficient anticoagulant proteins concentration

In the normal state there is a delicate balance between the proteins activating/participating the coagulation cascade and their inhibitors. Increased thrombin generation may result, as we presented above, from activation of the coagulation cascade due to higher concentrations or activities of the proteins included in the coagulation cascade. However, thrombin generation can also result from insufficient concentration or activity of anticoagulation proteins.

Tissue factor pathway inhibitor (TFPI), a glycoprotein comprising of three Kunitz domains (Broze et al., 1988) that are specific inhibitors of trypsin-like proteinases (Laskowski & Kato, 1980), is the main inhibitor of the extrinsic pathway of coagulation. TFPI inhibits thrombin generation through the inactivation of activated factor X and the factor VIIa/TF complex (Broze et al., 1988; Broze, Girard & Novotny, 1990). The mean maternal plasma concentrations of total TFPI increases during the first half of pregnancy, remains relatively constant in the second half (Sarig et al., 2005) and decreases during labor (Uszyński et al., 2001). There are two types of TFPI: (1) TFPI-1 is the more prevalent form in the non-pregnant state in the maternal circulation and can also be found in the fetal blood, platelets, endothelial cells and other organs (Edstrom, Calhoun & Christensen, 2000; Tay, Cheong & Boo, 2003); and (2) TFPI-2- the major form of TFPI in the placenta (Hubé et al., 2003; Iino, Foster & Kisiel, 1998; Sprecher et al., 1994; Udagawa et al., 1998), also known as Placental Protein 5 (PP5) (Kamei et al., 2001; Kisiel, Sprecher & Foster, 1994). During pregnancy, the maternal plasma concentration of TFPI-2 increases gradually, reaches a plateau at 36 weeks and subsides after delivery (Bützow et al., 1988; Chand, Foster & Kisiel, 2005; Seppälä, Wahlström & Bohn, 1979; Obiewke & Chard, 1981).

The overall balance between the concentration and activity of the coagulation factors and the anticoagulation proteins is one of the determining factors of thrombin generation. In the normal state, the immunoreactive concentrations of TFPI in the plasma are 500 to 1,000 times higher than that of TF (Shimura et al., 1997), suggesting that an excess of anticoagulant proteins closely control the coagulation cascade activity. The median maternal plasma TFPI concentration increases during preeclampsia (Erez et al., 2008; Abdel Gader et al., 2006), which is associated with an exaggerated maternal systemic inflammatory response. However, the increase in the median maternal TF plasma concentration is such that the overall balance between TF and its inhibitor is affected, leading to increased thrombin generation in this syndrome. In contrast to preeclampsia, maternal plasma TFPI concentration decreases in patients with PTL (Erez et al., 2010) and preterm PROM (Erez et al., 2008) regardless to the presence of intra-amniotic infection/inflammation, as well as in women with fetal demise (Erez et al., 2009), and does not change in mothers with SGA fetuses (Erez et al., 2008). Overall these findings suggest that the increased thrombin generation observed among these patients may derive not only from an increased activation of the hemostatic system, but also from insufficient anticoagulation, as reflected by the lower TFPI concentrations (Fig. 2).

Figure 2 (A) Comparison of median maternal plasma TF concentration between patients with normal pregnancy (n = 79), pre-eclampsia (n = 133), and women who delivered an SGAneonate (n = 61). (B) Comparison of median maternal plasma TFPI concentration between patients with normal pregnancy (n = 86), pre-eclampsia (n = 133), and women who delivered an SGA neonate (n = 61). (C) Comparison of maternal plasma TFPI/TF ratio between women with normal pregnancy (n = 79), pre-eclampsia (n = 133), and women who delivered an SGA neonate (n = 61). From Erez et al. (2008).

A possible explanation of the lower maternal plasma concentration observed in some of the obstetrical syndromes may be that during these syndromes there is a reduction in the placental production of TFPI (Hubé et al., 2003; Iino, Foster & Kisiel, 1998; Kamei et al., 2001; Abdel Gader et al., 2006) (mainly TFPI-2), contributing to the low maternal plasma concentrations detected in patients with PTL, in addition to the thrombin inhibitory effect to TFPI expression on endothelial cells, as above mentioned. Indeed, patients with vascular complications of pregnancy (preeclampsia, eclampsia, placental abruption, fetal growth restriction, and fetal demise) have a lower placental concentration of total TFPI, and TFPI mRNA expression than in women with normal pregnancies (Xiong et al., 2010; Aharon et al., 2005).

Other proteins implicated in the inhibitory control of the coagulation cascade are protein S, protein C and protein Z. Protein S is a cofactor to protein C in the inactivation of factors Va and VIIIa. This protein exists in two forms: a free form and a complex form bound to complement protein C4b-binding protein (C4BP). Only the free form is active (Castoldi & Hackeng, 2008). Protein S also acts as a TFPI cofactor, in the presence of weak procoagulant stimuli, by enhancing the interaction of TFPI with factor Xa while using Ca2+ and phospholipids in the process (Hackeng et al., 2006) without increasing inhibition of factor VIIa-TF by TFPI (Ndonwi & Broze, 2008). During pregnancy there is a physiologic change in the relationship between the bound and the free forms of protein S in the maternal plasma. The increase in C4BP during gestation reduces free protein S concentration in up to 55% of its value out of pregnant state, reaching its nadir at delivery. Of interest, cesarean delivery and infection exacerbate the reduction in free protein S concentrations (Bremme, 2003; Faught et al., 1995). Moreover, a functional protein S deficiency can explain a poor response to activated protein C (Dahlbäck, Carlsson & Svensson, 1993).

The association between the alteration of concentration and function of protein S and protein C in the great obstetrical syndromes is not completely clear. The evidence regarding the association of protein S and protein C deficiency and preeclampsia is controversial (Rodger et al., 2008; Yalinkaya et al., 2006).

While some reported an association between protein S deficiency and an increased risk for this syndrome (especially for early onset preeclampsia) (Rodger et al., 2008) others could not demonstrate this effect (Yalinkaya et al., 2006). There is some evidence regarding the relation of protein S deficiency and increased risk of stillbirth (Preston et al., 1996) and mid-trimester IUGR (Kupferminc et al., 2002). An increased risk of stillbirth has been reported in patients with protein S deficiency while the risk was not significantly increased in cases of protein C deficiency (Preston et al., 1996), and Kupferminc et al. (2002) found that protein S, but not protein C deficiency, was significantly associated with severe mid-trimester IUGR.

Protein Z, in complex with protein Z-dependent protease inhibitor (ZPI) (Fig. 3) (Han, Fiehler & Broze, 1998; Han et al., 1999; Han, Fiehler & Broze, 2000), acts as a physiologic inhibitor of activation of prothrombin by factor Xa. Protein Z is a vitamin K-dependent plasma glycoprotein (Yin et al., 2000) that is an essential cofactor for ZPI activity. In the absence of protein Z, the activity of ZPI is reduced by more than 1,000-fold (Han, Fiehler & Broze, 2000). Normal pregnancy is characterized by an increased plasma concentration of protein Z (Taylor et al., 1998), probably as a compensation for the increase of factor X concentration. Women with preterm labor without intra-amniotic infection or inflammation and those with vaginal bleeding who delivered preterm had a lower median maternal plasma protein Z concentration than women with a normal pregnancy and those with vaginal bleeding who delivered at term (Kusanovic et al., 2007). The changes of protein Z concentrations in other pregnancy complications are controversial. Some demonstrated that the median plasma concentration of protein Z in patients with preeclampsia, IUGR, and late fetal death were not significantly different than that of patients with a normal pregnancy (Bretelle et al., 2005). Others reported lower median maternal plasma protein Z concentrations in women with preeclampsia or pyelonephritis and higher proportion of protein Z deficiency (defined as protein Z plasma concentration below the 5th percentile) in patients with preeclampsia or fetal demise than in those with a normal pregnancy (Nien et al., 2008). Moreover, increased maternal plasma anti-protein Z antibodies concentrations were associated with SGA neonates, fetal demise and preeclampsia.

Figure 3 Factor X activation and protein Z/protein Z-dependent protease inhibitor (ZPI) inhibition of activated factor X.

(A) The formation of the complex of tissue factor (TF) and factor VIIa (FVIIa) at the site of injury and activation of extrinsic coagulation cascade. (B) Activation of circulating factor X by the TF/VIIa complex in the presence of exposed phospholipids and Ca2+. (C) Inhibition of factor Xa (FXa) by the protein Z/ZPI complex by binding to its active site. Modified from Broze (2001).

The information presented above suggests that it is not only the concentration of one coagulation factor or anticoagulation protein, but rather the overall balance between the coagulation factors and their inhibitors that increases thrombin generation in the great obstetrical syndromes. Indeed, although preterm labor was not associated with a significant change in the median maternal plasma TF concentration, the TFPI/TF ratio of these patients was lower than that of normal pregnant women, mainly due to decreased TFPI concentrations.

This observation was also reported in patients with preterm PROM (Erez et al., 2008), and those with preeclampsia (Erez et al., 2008). The lower TFPI/TF ratio in patients with preeclampsia occurs despite the increase in the median maternal plasma TFPI concentration observed in these patients. This suggests that the balance between TF and its natural inhibitor may better reflect the overall activity of the TF pathway of coagulation, than the individual concentrations of TF or TFPI.

Collectively, these observations suggest that our attention should be focused not only on the coagulation protein but also on their inhibitors since an imbalance between them may contribute to increased thrombin generation leading to the onset of the great obstetrical syndromes.

Changes in the feto-maternal interface

Normal placental development and the establishment of an adequate feto-maternal circulation are key points for a successful pregnancy. The networks of the placental vascular tree either on the maternal or fetal side are dynamic structures that can be substantially altered in cases of abnormal placentation and trophoblast invasion. The human trophoblast has properties of endothelial cells and can regulate the degree of activation of the coagulation cascade in the intervillous space (Sood et al., 2006; Sood et al., 2008). The villous trophoblasts express heparin sulfate, protein C and protein Z on their surface that serve as anticoagulant that sustain laminar blood flow through the intervillous space. On the other hand, unlike the endothelium of other organs, the trophoblast constantly presents the active placental isoform of TF on its surface (Sood et al., 2008; Lanir, Aharon & Brenner, 2003; Isermann et al., 2003; Aharon et al., 2004). This isoform has a higher affinity for factor VIIa (Butenas & Orfeo, 2007), which may lead to increased activation of the coagulation cascade. One of the leading pathological processes observed in all these syndromes is thrombosis and vascular abnormality of the placenta at the maternal–fetal interface. The incidence of these pathological processes varies among the different syndromes being more prevalent in preeclampsia, IUGR, and fetal demise than in PTL and preterm PROM (Schjetlein et al., 1999; Chaiworapongsa et al., 2002; Erez et al., 2009; Elovitz, Baron & Phillippe, 2001).

Placental pathology in the great obstetrical syndromes

There is a range of placental vascular and thrombotic lesions that are being observed in placentas of patients with pregnancy complications. Thrombotic events of placental vessels can cause an impairment of placental perfusion, leading to fetal growth restriction (FGR), preeclampsia and fetal death as well as in some extents to PTL and preterm PROM (Midderdorp, 2007; Martinelli et al., 2001). The frequency of the specific vascular placental lesions varies among these obstetrical syndromes (Kovo, Schreiber & Bar, 2013).

Placental vascular lesions are divided into maternal or fetal vascular origin (Figs. 4 and 5) (Redline et al., 2005; Bar et al., 2012). Lesions of the maternal vascular compartment include placental marginal and retro-placental hemorrhages, lesions related to maternal underperfusion (acute atherosis and mural hypertrophy, increased syncytial knots, villous agglutination, increased intervillous fibrin deposition, villous infarcts) (Redline et al., 2005). Placental fetal vascular obstructive lesions are the result of stasis, hypercoagulability and vascular damage within the fetal circulation of the placenta. Placental fetal vascular abnormalities include: cord-related abnormalities (as torsion of cord, over-coiling, strictures and tight knots (Cromi et al., 2005)) and vascular lesions consistent with fetal thrombo-occlusive disease (thrombosis of the chorionic plate and stem villous vessels, fibrotic, hypo-vascular and avascular villi (Redline et al., 2005)). In addition, villitis of unknown etiology or chronic villitis, defined as lymphohistiocytic inflammation localized to the stroma of terminal villi but often extending to the small vessels of upstream villi is also associated with obliterative fetal vasculopathy (Redline et al., 2005) (Figs. 4 and 5).

Figure 4 Histologic features of maternal vessels and implantation site reaction patterns.

(A) Acute atherosis of maternal arterioles in the placental membranes: a cluster of decidual arterioles shows varying stages of fibrinoid necrosis. The vessel at the upper right shows full histologic expression with dark homogenous fibrinoid replacement of the vessel wall accompanied by occasional foamy macrophages ([original magnification is indicated for all panels] X 20). (B) Mural hypertrophy of decidual arterioles in the placental membranes: a cluster of arterioles shows medial hypertrophy with the vessel wall occupying greater than one third of total vessel diameter (X 10). (C) Muscularized basal plate arteries with accompanying implantation site abnormalities: maternal spiral arteries in the basal plate lack normal trophoblast remodeling and retain their pre-pregnancy muscular media. Clusters of immature intermediate trophoblast and increased placental giant cells are seen above and below the muscular arteries, respectively (X 10). (D) Acute atherosis of muscularized basal plate arteries with accompanying implantation site abnormalities: three cross sections of a basal plate artery are seen. The two on the left show persistence of the muscular media while the one on the right has undergone fibrinoid necrosis of the media with foamy macrophages (acute atherosis). Clusters of immature intermediate trophoblast are also seen overlying the arteries (X 4). (E) Immature intermediate trophoblast: clusters of abnormally small intermediate trophoblast with focal vacuolation are surrounded by an excessive amount of basal plate fibrin. Increased placental site giant cells are also seen at the lower margin (X 10). (F) Increased placental site giant cells: numerous multinucleate placental site giant cells, not usually seen in the delivered placenta, are scattered in loose decidual tissue which is devoid of normal intermediate trophoblast and fibrinoid (X 10). From Redline et al. (2004).

Figure 5 Histologic features of villous and intervillous lesions.

(A) Increased syncytial knots: aggregates of syncytiotrophoblast nuclei cluster at one or more poles of distal villi in the vicinity of larger stem villi (arrowhead) at the periphery of the lobule ([original magnification is indicated for all panels] X 10). (B) Villous agglutination: clusters of degenerating distal villi are adherent to one another and focally enmeshed in fibrin (X 4). (C) Distal villous hypoplasia: a long, thin, non-branching stem villus is surrounded by a markedly reduced number of small hypoplastic distal villi (X 10). (D) Increased intervillous fibrin: stem villi are surrounded by a mantle of fibrin-type fibrinoid that does not extend to distal villi at the center of the lobule (X 2). (E) Nodular intervillous (and intravillous) fibrin: small aggregates of intervillous fibrin adhere to, and are focally reepithelialized by, distal villous trophoblast (X 20). (F) Increased intervillous fibrin with intermediate trophoblast (X-cells): stem and distal villi are enmeshed in a matrix of fibrin and fibrinoid containing prominent intermediate trophoblast (arrowhead) (X 10). From Redline et al. (2004).

Preeclampsia. The classic example for an association between obstetrical syndromes and vascular placental lesions is preeclampsia. Women who develop preeclampsia have an increased rate of abnormalities of the maternal side of the placental circulation and maternal underperfusion (Salafia et al., 1998; Roberts & Post, 2008). The frequency of these lesions is inversely related to the gestational age in which the hypertensive disorder was diagnosed. The earlier the development of hypertension/preeclampsia the more severe the vascular lesions are (Mayhew et al., 2003; Ogge et al., 2011). Moreover, Kovo et al. (2010) reported that the presence of fetal growth restriction in women with preeclampsia also increases the frequency of fetal vascular lesions. Indeed, patients with early-onset preeclampsia complicated by FGR had a higher rate of fetal vascular supply lesions consistent with fetal thrombo-occlusive disease than women with early onset disease without FGR (Kovo et al., 2010).

An assessment of the pathologic changes in placental hemostatic system has been performed in patients with preeclampsia. Teng et al. (2010) studied TF and TFPI placental levels in pregnant patients with preeclampsia, compared to normal pregnancies. They found increased TF placental expression and a reduced expression of TFPI-1 and TFPI-2, with a significant correlation between the levels of TF and TFPI-2 between maternal plasma and placenta.

Fetal growth restriction. Placentas from pregnancies complicated by FGR are smaller and have significantly increased maternal and fetal vascular lesions compared to placentas from normal pregnancies with appropriate for gestational age neonates (AGA) (Rerdline, 2008; Salafia et al., 1995). Maternal vascular lesions were detected in about 50% of placentas from pregnancies complicated with FGR at term, compared to only 20% in normal pregnancies, while fetal vascular lesions were observed in 11% of FGR pregnancies compared to only 4% in placentas from normal pregnancies (Kovo et al., 2010).

Placentas from normotensive pregnancies complicated by early onset FGR (<34 weeks of gestation) had a higher rate of low placental weight (<10th percentile) and maternal underperfusion, as compared to placentas of women who delivered AGA neonates ≤34 weeks of gestation (Rerdline, 2008). Of interest, placentas from the late onset FGR group (after 34 weeks of gestation), in addition to the high maternal vascular abnormalities, show also more fetal vascular abnormalities, compared with AGA controls who delivered >34 weeks (Kovo et al., 2012).

Fetal demise. Placental disease has been recognized as an important contributor to unexplained fetal demise. Fetal vascular abnormalities (Kovo, Schreiber & Bar, 2013) are extensively involved in early and late fetal death rather than maternal vascular lesions. In fetal death occurring prior to 34 weeks, an earlier and extended insult in the placental development occurs. On the other hand, late fetal demise is an unpredicted event that is mostly characterized by non-thrombotic cord related lesions and less placental vascular compromise (Bar et al., 2012).

Preterm labor and preterm PROM. Placental studies in PTL demonstrated a combination of inflammatory and vascular lesions. PTL is generally attributed to an inflammatory response involving the bacterial induction of cytokine and prostanoid production (Romero et al., 2006). Finding of histological chorioamnionitis in PTL (Arias et al., 1993) has established infection and inflammation as a causative factor of preterm birth, moreover, noninfectious trigger may also contribute to the development of preterm labor and in some instances may be evident by placental sterile inflammatory response (Nath et al., 2007). In addition, isolated placental vascular lesions, mostly of maternal supply, were reported in 20% of cases of PTL and an additional 20% had combined inflammatory and vascular lesions. Moreover, there are consistent reports describing increased rate of failure of transformation of the spiral arteries in women with preterm labor without intrauterine infection/inflammation and in those with preterm PROM than in women with normal pregnancies (Kim et al., 2002). Such findings imply that an inadequate uteroplacental blood flow due to abnormal placentation plays an important role in pathogenesis of preterm parturition (Kim et al., 2002; Salafia & Vogel, 1991).

Collectively, placental vascular lesions were reported in all the great obstetrical syndromes. The severity of these lesions is associated with the timing of diagnosis of the disease. The more severe the vascular injury, the more likely these complications will become clinically evident prior to 34 weeks of gestation. Of interest, vascular lesions often come along with evidence of acute inflammation or lesions associated with chronic inflammatory processes, suggesting that sometimes more than one mechanism is involved in the development of a specific obstetrical syndrome.

Hemostatic changes in the amniotic fluid of women with obstetrical syndromes

During normal pregnancy, there is an increase in the amniotic fluid TF concentration (de Boer et al., 1989; Uszyński et al., 2001; Lockwood et al., 1991; Omsjø et al., 1985; Creter, 1977). In order to demonstrate the association of hemostatic changes and the development of obstetrical complications, Erez et al. (2009) studied the changes in the intra-amniotic concentration of TAT III complexes, as well as TF concentration and activity, in cases of fetal demise and in normal pregnancies.

Patients with a fetal demise had higher median amniotic fluid TF concentration and activity than those with normal pregnancies. Moreover, among patients with fetal demise there was a significant correlation (Fig. 6) between the amniotic fluid TF concentrations and activity (r = 0.88, P < 0.0001). The median amniotic fluid TAT III complexes concentration did not differ significantly between the groups (normal pregnancy: median: 66.3 mg/L, range 11.4–2,265.4 vs. fetal demise median: 59.3 mg/L, range: 13.6–15,425.3; P = 0.7). In their study, the median amniotic fluid TF concentration in normal pregnant women was 10 fold higher than in maternal plasma.

Figure 6 (A) Amniotic fluid tissue factor concentration among women with normal pregnancies (median 3,710.4 pg/mL, range 2,198.8–6,268) and patients with a fetal demise (median 8,535.4 pg/mL, range 2,208.2–1,25,990.0). (B) Amniotic fluid tissue factor activity among women with normal pregnancies (median 28.4 pM, range 10.2–84.9) and patients with a fetal demise (median 81.6 pM, range 7.2–1,603.4). From Erez et al. (2009).

The changes in amniotic fluid thrombin generation were reported also in women with preterm parturition. Indeed, intra-amniotic infection and/or inflammation is associated with an increased amniotic fluid TAT III complexes (Fig. 7). This is important since it represents an increased thrombin generation in the amniotic cavity during infection and/or inflammation that may contribute to uterine contractility and the development of preterm birth (Stephenson et al., 2005). Of interest, elevated intra-amniotic TAT III concentrations were associated with a shorter amniocentesis to delivery interval and an earlier gestational age at delivery only in patients with preterm labor without intra-amniotic infection or inflammation (Stephenson et al., 2005). This observation suggests that in a subset of patients with preterm labor, activation of the coagulation system can generate preterm parturition and delivery; while in those with intra-amniotic infection and/or inflammation the activation of the coagulation and thrombin generation is a byproduct of the inflammatory process leading to preterm birth.

Figure 7 Maternal plasma TAT III concentration in women with preterm labor (PTL) and those with a normal pregnancy. From Chaiworapongsa et al. (2002).

This represents evidence of the activation and propagation of coagulation cascade, being thrombin generation the witness of the former mechanisms and the inhibitor of the initiation step (Erez et al., 2009).

Conclusion

The evidence presented herein suggests a role for increased thrombin generation and vascular placental lesions in the pathogenesis of the great obstetrical syndromes. This process can be the result of the contribution of procoagulant and vascular abnormalities as well as inflammatory and infectious mechanisms, representing the starting point for pregnancy complications based on vascular disease.

As presented, these changes affect the mother, the placenta, the membranes and the amniotic fluid. Moreover, preliminary evidence suggests that some of the changes in the hemostatic system in the mother and in the amniotic fluid predate the clinical presentation of the disease. This means that better understanding of the vascular and coagulation changes associated with the great obstetrical syndromes may assist us in earlier detection and the development or introduction of therapeutic modalities for these syndromes.

Additional Information and Declarations

Competing Interests

Author Contributions

Offer Erez is an Academic Editor for PeerJ.

Salvatore Andrea Mastrolia, Moshe Mazor, Giuseppe Loverro, Vered Klaitman and Offer Erez wrote the paper, prepared figures and/or tables, reviewed drafts of the paper.

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
