# Peer review of "Placental vascular pathology and increased thrombin generation as mechanisms of disease in obstetrical syndromes"

_PeerJ, doi:10.7717/peerj.653_

## Round 0.1 · original submission · Minor Revisions

One reviewer found your manuscript not ideally suited for a broad audience, but informative for a specialized one. In that regard, your revision should be mainly focused on generalizing some aspects of the clinical and scientific content. For the remainder, mainly minor suggestions were given by the reviewers. These can be found below and should be addressed appropriately.

Reviewer 1 ·

Basic reporting

NA

Experimental design

NA

Validity of the findings

NA

Additional comments

I read with interest this review of coagulation changes during pregnancy that may cause severe pregnancy complications if the balance between coagulation and anti-coagulation is disturbed.

- Is the topic of relevance to a wide enough audience (i.e. does this topic need a literature review)

This review is informative for a specialized public. Obstetricians, gynaecologists, scientists working in the field of coagulation find a good overview of what is currently known (or not known) about the pathogenesis of obstetrical complications.

- Is the topic of the article adequately described / introduced

Yes. This review describes clearly the topic.

- Are the references properly cited? Yes

- Are the references up-to-date? Yes

- Does the author comprehensively survey the complete landscape of the topic? Yes
Are there any gaps in coverage? This is a very broad topic. The role of inflammation could be more extensively reported; however, that could make the manuscript to complex.

- Is the manuscript understandable by a non-expert? (i.e. a practitioner, but someone who isn’t intimately involved in this field)
See my first remark. To my opinion, this is for a public of specialized readers.

- If the authors cite their own work, do they do so objectively?
Yes

- Are major areas of agreement and disagreement in the literature included and discussed objectively? Yes

- Does it have adequate illustrations or drawings?
Figure 4 is not clear

Other remarks:
Use of abbreviations and full text should be checked all over the manuscript; some examples:
- Line 103: FGR: first abbreviation, in full
- Line 137: tissue factor: use abbreviation
- Line 139: SGA: first abbreviation, in full

·

Basic reporting

No comments

Experimental design

No comments

Validity of the findings

No comments

Additional comments

This review article describes the changes in the hemostatic system during normal and complicated pregnancies in maternal blood, maternal-fetal interface and amniotic fluid, as well as the contribution of thrombosis and vascular pathology to the development of the great obstetrical syndrome. In general, well written and focused description. Tables and Figures are informative. The references are comprehensive and give credit to other authors writing on this subject.

I have only minor comments for the authors:
1) Line No 67-68,
“Indeed, there is a substantial increase in tissue factor (TF) concentrations in the decidua and myometrium, as well as preventing placental abruption”… Explain the details and add reference
2) Line 268 “Placental lesions are divided into….origin (figure 1-2)” Sentence does not match with figure 1-2. It could be figure 4-5.
3) Line 654. Figure 1. Figure legend and Figure does not match.
4) Check spelling

---

## Round 0.2 · accepted · Accept

I have no further comments and congratulate you with this PeerJ pioneering step of writing reviews on specific topics!